# Patterns of healthcare utilisation in children and young people: a retrospective cohort study using routinely collected healthcare data in Northwest London

Thomas Beaney ![ORCID] [1,2] Jonathan Clarke ![ORCID] [3,4] Thomas Woodcock,[1,2] Rachel McCarthy,[5] Kavitha Saravanakumar,[5] Mauricio Barahona ![ORCID] [3,4] Mitch Blair,[1,2] Dougal S Hargreaves[1,2]

For numbered affiliations see end of article.

**Correspondence to**
Dr Thomas Beaney;
thomas.beaney@imperial.ac.uk

## ABSTRACT

**Objectives** With a growing role for health services in managing population health, there is a need for early identification of populations with high need. Segmentation approaches partition the population based on demographics, long-term conditions (LTCs) or healthcare utilisation but have mostly been applied to adults. Our study uses segmentation methods to distinguish patterns of healthcare utilisation in children and young people (CYP) and to explore predictors of segment membership.

**Design** A retrospective cohort study.

**Setting** Routinely collected primary and secondary healthcare data in Northwest London from the Discover database.

**Participants** 378 309 CYP aged 0–15 years registered to a general practice in Northwest London with 1 full year of follow-up.

**Primary and secondary outcome measures** Assignment of each participant to a segment defined by seven healthcare variables representing primary and secondary care attendances, and description of utilisation patterns by segment. Predictors of segment membership described by age, sex, ethnicity, deprivation and LTCs.

**Results** Participants were grouped into six segments based on healthcare utilisation. Three segments predominantly used primary care, two moderate utilisation segments differed in use of emergency or elective care, and a high utilisation segment, representing 16 632 (4.4%) children accounted for the highest mean presentations across all service types. The two smallest segments, representing 13.3% of the population, accounted for 62.5% of total costs. Younger age, residence in areas of higher deprivation and the presence of one or more LTCs were associated with membership of higher utilisation segments, but 75.0% of those in the highest utilisation segment had no LTC.

**Conclusions** This article identifies six segments of healthcare utilisation in CYP and predictors of segment membership. Demographics and LTCs may not explain utilisation patterns as strongly as in adults, which may limit the use of routine data in predicting utilisation and

## Strengths and limitations of this study

► Segmentation approaches have been widely used in the adult population for the purposes of population health management but are less widely used in children.

► This study uses routinely collected primary and secondary care data, including all eligible patients registered to general practices in Northwest London region.

► Using a data-driven approach with k-means clustering, we defined six distinct segments of the child population in Northwest London based on healthcare utilisation patterns.

► Age, sex, ethnicity, deprivation and number of long-term conditions (LTCs) were strong predictors of utilisation segment, but three-quarters of children in the highest utilisation segment had no LTC.

► Demographics and comorbidities may not capture the variation in healthcare utilisation to the same extent as in adults and further research is needed to identify whether additional factors in the electronic health record can predict utilisation in children.

suggest children have less well-defined trajectories of service use than adults.

## INTRODUCTION

There is growing recognition, both internationally and within the UK, of the role of the health service in managing population health.[1–3] Population health has become a core responsibility of NHS organisations and encompasses the activities that can be taken at a system level to improve physical and mental health and well-being, and reduce health inequalities for a whole population, rather than focusing solely on provision of services for individual patients.[4] Critical to

population health activities is the ability to identify and understand the needs of those for whom early intervention may improve outcomes, and thus reduce the need for more costly hospital-based care. The life course approach to prevention suggests that the long-term benefits of early intervention are greatest in children and young people (CYP), particularly in the first 1000 days of life, highlighting the need for population health approaches tailored toward younger people.[5]

Segmentation methods are an approach to categorising individuals with similar characteristics with the aim of identifying common care needs and designing services tailored to those in each segment to improve health outcomes and optimise healthcare utilisation.[6 7] A variety of approaches to population segmentation in health settings have been described, based mostly on demographic and medical factors and including both expert-driven and data-driven designs, or a mix of the two.[8 9] Segmenting on healthcare utilisation is an alternative, and included in some approaches, but only one previous segmentation method was based solely on utilisation.[10] Existing work on segmentation methods have applied to adults, who may represent a very different population to that of CYP in terms of chronic disease burden and patterns of healthcare utilisation, indicating a need for research focused on the CYP population.

In the adult population, healthcare utilisation and expenditure are closely associated with increasing age and comorbidities: in a recent study of healthcare costs in England, adults accounting for the top 5% of costs were significantly older and 85.5% had at least one long-term condition (LTC).[11] In contrast, in the CYP population, a higher proportion of children under 5 years of age accounted for the top 5% of costs (compared with those aged 5–9 years and 10–14 years), with only half of those in the high-cost group having no defined LTC.[12] These findings suggest that approaches to segmentation using age and LTC will only partially account for the utilisation and costs associated with care in CYP, and that other factors, including those available in electronic health records (EHRs) such as sex, ethnicity and deprivation, may also play a role.

Exploration of the link between modifiable and non-modifiable risk factors in relation to children's use of health services is key to developing our understanding of the utility and limitations of segmentations approaches, and the extent to which health need can be identified and intervened on pre-emptively. The aim of this article is to use a data-driven method to characterise different patterns of healthcare utilisation over the course of a year in the CYP population for Northwest London, using routinely available healthcare data. We then describe the population characteristics of the resulting segments, by age, sex, ethnicity, LTC burden and deprivation in order to identify population groups with different patterns of healthcare utilisation.

## METHODS

### Data source and cohort design

Data for this study used the Discover platform to access the Whole Systems Integrated Care (WSIC) database. WSIC holds the records of 2.4 million patients registered to general practices (GPs) within Northwest London, representing 95% of the registered population and comprising both primary and secondary care data.[13] Primary care data are extracted directly from the EHR, and are linked to secondary care data from the Secondary Uses Service dataset of accident and emergency (A&E) attendances, inpatient admissions (including both elective and emergency admissions) and outpatient attendances covering all hospital providers in England.

A retrospective cohort was constructed of all CYP aged 0–15 years who were registered as of 1 March 2019 with a full year (365 days) of follow-up until 29 February 2020. These dates were chosen to provide contemporaneous data while minimising the overlap with the COVID-19 pandemic, due to expected changes in usual healthcare utilisation patterns. For children who had died over the study time period, 365 days of follow-up were included prior to the date of death to ensure a comparable time period, allowing an earliest study start date of 1 March 2018.

### Healthcare utilisation, cost and LTCs

Seven healthcare utilisation variables were chosen, for which attendance is robustly recorded in the clinical record and summed across the year

1. GP attendances (including all consultation and staff types)
2. A&E attendances
3. Outpatient attendances
4. Elective hospital admissions
5. Total length of stay from elective admissions (days)
6. Emergency hospital admissions
7. Total length of stay from emergency admissions (days)

Estimated healthcare costs for primary care were derived from Curtis and Burns's study,[14] assuming average consultation times and including direct care staff costs.[14] Primary care prescription costs were not included. Secondary care costs are included directly in Discover database based on the total National Tariff Payment System cost associated with each attendance or admission.[15] Where these were missing, costs were estimated based on Curtis and Burns's study[14] (for paediatric outpatient attendances) or the National Schedule of NHS Costs (A&E attendances and emergency/elective admissions).[16] Comorbidities were selected from among 41 LTCs defined in Discover database. A subset of 17 of these chronic conditions identified in previous literature as relevant to children was included[17] (online supplemental table S1), supplemented by additional SNOMED code sets for relevant conditions not included in Discover database (online supplemental table S2)[18] identified from the entirety of the patient primary care record. Secondary care diagnostic codes were not included in order not to bias number of conditions in

favour of those presenting to secondary care, although they may be coded in the primary care record. Further details of the healthcare utilisation variables and LTCs are given in the online supplemental file 1 (see pages 2–4).

## Cluster analysis

Two methods were considered for cluster analysis, which have been widely used in the segmentation literature: k-means and hierarchical agglomerative clustering, following the approach used by Vuik *et al*.[9 10] Given our population approach, use of the entire dataset was preferrable. Hierarchical methods are computationally intensive and may not run on datasets with several hundreds of thousands of samples and so k-means was chosen as the primary clustering method with a sensitivity compared with Ward's hierarchical agglomerative clustering on a subset of data.[10] Distributions of each utilisation measure were examined using histograms, were log-transformed to reduce skew and rescaled using minimum–maximum scaling. K-means clustering was performed, partitioning the cohort using Euclidean distance on the seven transformed and scaled utilisation variables. The algorithm was iterated 10 000 times with different randomly placed initial cluster centroids, and the iteration was selected which minimised the within-cluster sum of squares.[19] A range of models with a number of k clusters from 2 to 14 was compared using Davies-Bouldin (D-B) and Calinski-Harabasz (C-H) scores.[20 21] Models with low D-B scores and high C-H scores were examined further: (a) confusion matrices were constructed to compare cluster assignment in the model with k clusters against the model with k+1 clusters and (b) mean values for each utilisation variable were calculated for each cluster in each model. Further details of the clustering approach are given in online supplemental file 1 (see pages 5–17).

## Statistical analysis

For the chosen clustering model, the resulting population segments were described by mean utilisation and characterised by age, sex, ethnicity, deprivation and LTC. Deprivation was calculated using quintiles of the Index of Multiple Deprivation (IMD) linked to patient postcode, with 1 representing the most deprived and 5 indicating the least deprived quintile.[22] Univariable associations were compared using $\chi^2$ tests. Adjusted multinomial logistic regression was used to identify characteristics associated with assignment to a particular segment relative to the lowest utilisation segment, including age, sex, ethnicity, deprivation and LTC in the model. The association between death and segment assignment was examined using univariable logistic regression. Missing data were low for gender (<0.1%) and IMD quintile (4.3%), but ethnicity was missing in 25.4% of cases. A sensitivity analysis of the multinomial model was carried out with the exclusion of ethnicity as a predictor.

Data were extracted from Discover using Microsoft SQL Server Management Studio 2012. Stata V.16.1 (StataCorp) was used for data manipulation, characterisation of segments and multinomial regression. Cluster analysis used Python V.3.6.8, Pandas V.1.2.0 and Scikit-learn V.0.23.2.[23]

## Patient and public involvement

Patients or the public were not involved in this study.

# RESULTS

## Population characteristics

A total of 429 496 CYP aged 0–15 years registered to GPs in Northwest London were identified in Discover. Of these, 378 266 (88.1%) participants were currently registered and had a full year of follow-up between 1 March 2019 and 29 February 2020. An additional 43 children (0.01%) died during this time period, and a full year of follow-up was included preceding the date of death. A total of 378 309 participants were, therefore, included in the cohort.

Characteristics of the population are given in table 1. 51.3% of children were male, and 48.7% female with a mean (SD) age of 7.4 years (4.6 years). Ethnicity was documented for 74.6% of participants, with the most common ethnic category being Asian or Asian British (24.5%), followed by 22.0% white, 11.0% mixed, 10.1% other ethnic groups and 6.9% black. There was a higher proportion of the population falling into more deprived areas by IMD quintile, with 16.0% in the most deprived, 32.0% in the second most deprived and only 7.1% in the least deprived quintile. A total of 35 946 participants (9.5%) were recorded as having one or more LTCs, with 32 963 (91.7%) of these having one LTC and 2983 (8.3%) having two or more LTCs. Asthma was the most commonly recorded chronic condition, with 21 659 prevalent cases (5.7%), followed by obesity (6995; 1.9%), mental health problems (2893; 0.8%), learning disabilities (1607; 0.4%) and chronic heart conditions (1523; 0.4%).

## Healthcare utilisation

A total of 264 755 (70.0%) CYP had at least one attendance to their GP over the year, with a mean (SD) number of attendances of 2.8 (3.9), and 19 990 (5.3%) children presenting 10 or more times (table 2). A total of 136 424 children (36.1%) were matched with one or more records of any secondary care attendance, with 23.5% attending A&E and 21.2% attending outpatients at least once over the year. A total of 9461 children (2.5%) had one or more elective admissions, with a mean length of stay of 0.3 days per admission. 87.2% of those with at least one elective admission had no overnight admissions over the year. A total of 12 078 children (3.2%) had at least one emergency admission, with a mean length of stay of 1.6 days per admission. Of those children admitted at least once as an emergency, 41.9% had no overnight admissions over the year.

**Table 1** Participant demographics and LTCs at recruitment for all 378 309 participants

| Participant characteristic | | Total | Percentage |
|---|---|---|---|
| Gender | Female | 184 408 | 48.7% |
| | Male | 193 895 | 51.3% |
| | Unknown | 6 | <0.1% |
| Age (years) | Mean (SD) | 7.4 (4.6) | |
| | Median (IQR) | 7 (3–11) | |
| | Under 1 | 28 304 | 7.5% |
| | 1–2 | 44 590 | 11.8% |
| | 3–4 | 45 559 | 12.0% |
| | 5–10 | 148 025 | 39.1% |
| | 11–15 | 111 831 | 29.6% |
| Ethnicity | Asian or Asian British | 92 704 | 24.5% |
| | White | 83 406 | 22.0% |
| | Mixed/multiple ethnic groups | 41 727 | 11.0% |
| | Other ethnic groups | 38 279 | 10.1% |
| | Black or black British | 25 959 | 6.9% |
| | Unknown | 96 234 | 25.4% |
| IMD quintile | 1 (most deprived) | 60 506 | 16.0% |
| | 2 | 121 174 | 32.0% |
| | 3 | 96 213 | 25.4% |
| | 4 | 57 445 | 15.2% |
| | 5 (least deprived) | 26 713 | 7.1% |
| | Unknown | 16 258 | 4.3% |
| Number of LTCs | 0 | 341 911 | 90.4% |
| | 1 | 32 963 | 8.7% |
| | 2 | 2724 | 0.7% |
| | 3 | 235 | 0.1% |
| | 4 or more | 24 | <0.1% |
| | Missing data | 452 | 0.1% |
| LTCs* | Asthma | 21 659 | 5.7% |
| | Obesity | 6995 | 1.9% |
| | Mental health problem | 2893 | 0.8% |
| | Learning disability | 1607 | 0.4% |
| | Chronic heart disease | 1523 | 0.4% |
| | Chronic neurological disease and epilepsy | 1309 | 0.3% |
| | Hypothyroidism | 1088 | 0.3% |
| | Cancer and immunosuppression | 1061 | 0.3% |
| | Diabetes | 775 | 0.2% |
| | Chronic respiratory disease (non-asthma) | 126 | <0.1% |
| | Chronic kidney disease | 126 | <0.1% |
| | Palliative care | 36 | <0.1% |
| | Rheumatoid arthritis | 18 | <0.1% |
| Total participants | | 378 309 | |

*Proportion of those with non-missing data on LTC.
IMD, Index of Multiple Deprivation; LTCs, long-term conditions.

**Table 2** Healthcare utilisation over 1 year for all 378 309 participants

| Healthcare utilisation | | Total | Percentage |
|---|---|---|---|
| GP attendances | Mean (SD) | 2.8 (3.9) | |
| | Median (IQR) | 2 (0–4) | |
| | 0 | 113 554 | 30.0% |
| | 1 | 70 666 | 18.7% |
| | 2–4 | 116 328 | 30.7% |
| | 5–9 | 57 771 | 15.3% |
| | 10+ | 19 990 | 5.3% |
| A&E attendances | Mean (SD) | 0.4 (1.0) | |
| | Median (IQR) | 0 (0–0) | |
| | 0 | 289 316 | 76.5% |
| | 1 | 51 972 | 13.7% |
| | 2–4 | 33 022 | 8.7% |
| | 5–9 | 3724 | 1.0% |
| | 10+ | 275 | 0.1% |
| Outpatient attendances | Mean (SD) | 0.7 (2.3) | |
| | Median (IQR) | 0 (0–0) | |
| | 0 | 298 289 | 78.8% |
| | 1 | 28 437 | 7.5% |
| | 2–4 | 35 014 | 9.3% |
| | 5–9 | 12 448 | 3.3% |
| | 10+ | 4121 | 1.1% |
| Elective admissions | Mean (SD) | 0.04 (0.40) | |
| | Median (IQR) | 0 (0–0) | |
| | 0 | 368 848 | 97.5% |
| | 1–2 | 7498 | 2.0% |
| | 3–4 | 1709 | 0.5% |
| | 5–9 | 158 | <0.1% |
| | 10+ | 96 | <0.1% |
| Total elective length of stay (days)† | Mean (SD) | 0.24 (0.98) | |
| | Median (IQR) | 0 (0–0) | |
| | 0 | 8248 | 87.2% |
| | 1 | 822 | 8.7% |
| | 2–4 | 256 | 2.7% |
| | 5–9 | 76 | 0.8% |
| | 10+ | 59 | 0.6% |
| Emergency admissions | Mean (SD) | 0.04 (0.28) | |
| | Median (IQR) | 0 (0–0) | |
| | 0 | 366 231 | 96.8% |
| | 1–2 | 10 054 | 2.7% |
| | 3–4 | 1902 | 0.5% |
| | 5–9 | 108 | <0.1% |
| | 10+ | 14 | <0.1% |

Continued

**Table 2** Continued

| Healthcare utilisation | | Total | Percentage |
|---|---|---|---|
| Total emergency length of stay (days)† | Mean (SD) | 1.6 (4.2) | |
| | Median (IQR) | 1 (0–2) | |
| | 0 | 5058 | 41.9% |
| | 1–2 | 3778 | 31.3% |
| | 3–4 | 2302 | 19.1% |
| | 5–9 | 647 | 5.4% |
| | 10–14 | 293 | 2.4% |
| Total participants | | 378 309 | |

*Proportion of those with non-missing data on LTC.
†Percentages of those admitted at least once.
A&E, accident and emergency; GP, general practice; LTC, long-term condition.

## Model selection

Comparison of D-B and C-H scores for a range of values of k clusters from 2 to 14 from k-means clustering on the 7 health utilisation variables identified an optimal range from 4 to 8 clusters for further exploration (online supplemental table S3). Mean utilisation measures were compared for each of the clusters in the 4-cluster to 8-cluster models (online supplemental figures S1-S5) and confusion matrices compared assignment to segments between clusters with increasing number of k clusters (online supplemental figures S6-S9). Membership of clusters within the 6-cluster model was relatively consistent with respect to membership of clusters in the 5-cluster and 7-cluster models. Additional increases in number of clusters to 7 or 8 did not add to the clinical utility given the additional complexity of interpretation.

Hierarchical agglomerative clustering using Ward's method was feasible on sub-samples of up to 50 000 participants. Evaluation of cluster membership on 10 different random draws of data produced a mean V-measure score (a combination of homogeneity and completeness of cluster assignment) of 0.85 suggesting clustering was sensitive to sampling (online supplemental file 1, see pages 6–7).[24] Clusters were characterised for the 6-cluster model using Ward's method and were broadly similar to those from the 6-cluster k-means model (online supplemental figure S10 and table S4). Descriptively, both the 6-cluster k-means model and the 6-cluster hierarchical model partitioned according to the same variables, differing only in the magnitude of mean presentations, while the 'very low' utilisation group became a 'no-utilisation' group. Given the focus on describing the whole population, rather than a subsample, the 6-cluster k-means model was selected for further description.

## Utilisation by segment

For the chosen 6-cluster model, the resulting population segments based on healthcare utilisation were described. Most patients fell into either a very low utilisation segment (I; 29.3%) with almost no healthcare utilisation, or into a

low utilisation segment (II; 26.0%) attending a GP with a mean (SD) of 1.4 (0.5) times in a year with very little additional utilisation (table 3; medians (IQR) given in (online supplemental table S5). Three smaller moderate utilisation segments were identified, with above average primary care utilisation. The largest of these (III) used primary care with a mean (SD) of 5.2 (3.1) times per year, but had little other service use. The two remaining moderate utilisation segments were characterised by either above average elective care (IV; outpatient and elective admissions) or above average emergency care (V; A&E attendances and emergency admissions), attending primary care with a mean (SD) of 4.2 (3.7) or 5.0 (3.7) times, respectively.

A high utilisation segment (VI) was also identified, accounting for 16 632 children (4.4%), with the highest mean utilisation across all services (table 2). Over 50% of all emergency admissions in the population, and total length of stay for elective and emergency admissions were accounted for by this group. Figure 1A shows the distribution of total activity across the seven utilisation variables for each segment. Figure 1B shows the distribution of total activity if all segments were of equal size, denoting the relative share of activity for a given individual within each segment.

### Costs per segment

Total healthcare costs for the cohort amounted to £140 million over the year with a mean of £372 per child, of which £92 was accounted for by primary care, and £280 by secondary care costs. Higher utilisation segments accounted for larger per capita costs and overall costs (online supplemental table S6). Mean per capita costs ranged from £32 in segment I to £1222 in segment IV and £2807 in segment VI. The lowest utilisation segment (I) represented 29.3% of the population but only 2.5% of total costs, and the highest utilisation segment (VI) for 4.4% of the population but 33.2% of total costs. Segment IV also accounted for 29.2% of total costs. Costs were not evenly distributed across care type: segment V accounted for 55.4% of A&E costs but only 19.4% of total costs and segment III represented the largest proportion of GP costs (33.3%) but only 10.4% of total costs.

### Predictors of segment membership

Characteristics of each segment were described by age, sex, ethnicity, IMD quintile and LTCs (online supplemental table S7). The gender balance differed across segments (p<0.001 from $\chi^2$ test) with segments IV–VI containing a higher proportion of males than females (54.0%, 53.2% and 56.0%, respectively). Higher utilisation segments tended to include younger children (p<0.001), with 17.3% and 17.5% aged under 1 year in segments V and VI, respectively, compared with only 1.9% aged under 1 year in segment I. There was a strong association with deprivation (p<0.001) with 20.4% in the most deprived quintile and 5.6% in the least deprived in segment VI, compared with 13.8% in the most deprived

**Table 3** Mean presentations/length of stay over 1 year for each segment from 6-cluster k-means model (green indicates lower than average utilisation and red indicates higher than average utilisation)

| Segment | Size | Percentage | GP | Outpatient | A&E | Elective admissions | Elective length of stay | Emergency admissions | Emergency length of stay |
|---|---|---|---|---|---|---|---|---|---|
| I: very low | 110869 | 29.3% | 0.00 | 0.09 | 0.07 | 0.01 | 0.00 | 0.00 | 0.00 |
| II: low; GP | 98293 | 26.0% | 1.36 | 0.06 | 0.07 | 0.01 | 0.00 | 0.00 | 0.00 |
| III: moderate; GP | 67358 | 17.8% | 5.19 | 0.14 | 0.00 | 0.01 | 0.00 | 0.00 | 0.00 |
| IV: moderate; GP, OP and elective | 33633 | 8.9% | 4.23 | 4.47 | 0.22 | 0.15 | 0.03 | 0.01 | 0.01 |
| V: moderate; GP and emergency | 51524 | 13.6% | 4.97 | 0.17 | 1.75 | 0.02 | 0.00 | 0.11 | 0.08 |
| VI: high; all services | 16632 | 4.4% | 10.14 | 5.01 | 2.78 | 0.32 | 0.14 | 0.52 | 0.89 |
| Total | 378309 | | 2.78 | 0.71 | 0.42 | 0.04 | 0.01 | 0.04 | 0.05 |

A&E, accident and emergency; GP, general practice; OP, Outpatient.

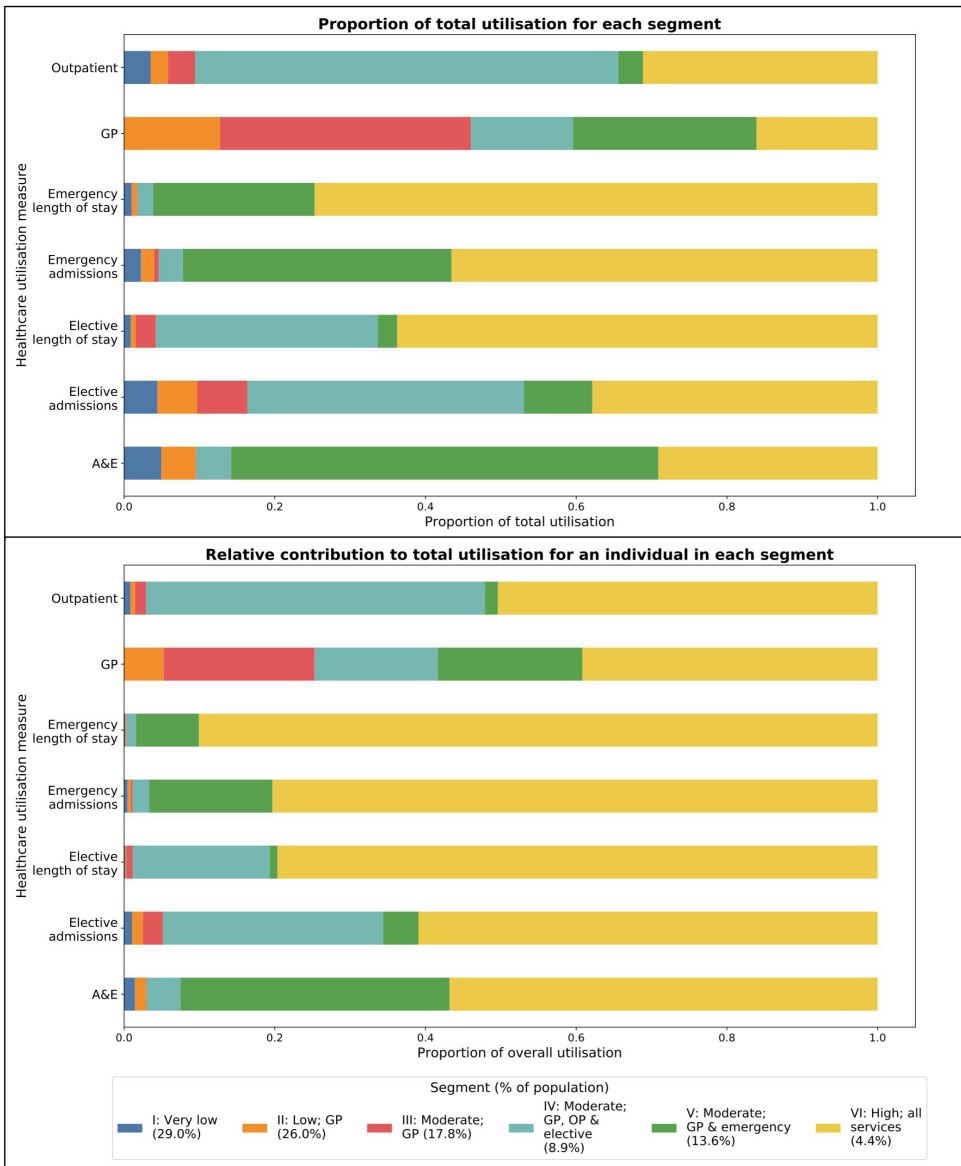

**Figure 1** (A) Proportion of total activity for each healthcare utilisation variables accounted for by each segment. (B) Relative contribution to total activity for each healthcare utilisation variable accounted for by a single individual in each segment assuming equal segment size. A&E, accident and emergency; GP, general practice; OP, outpatient.

and 7.5% in the least deprived in segment I. The proportion of the population with one or more LTCs increased across segments with increased utilisation (p<0.001), with only 3.6% in segment I, and 25.0% in segment VI. Children in the highest utilisation segment were significantly more likely to have died than those in the lowest utilisation segment (OR: 21.4, 95% CI: 7.8 to 58.3), although 62.8% of all deaths still occurred outside the highest utilisation segment and 34.9% of deaths occurred in segments I–III.

Adjusted multinomial logistic regression was conducted to identify the predictors of segment membership. Relative risk ratios (RRR) are presented in table 4. The relative risk of males being in segment VI compared with segment I was 17% (95% CI: 12% to 21%) higher compared with females after adjusting for age, ethnicity, deprivation and LTC. Age was a strong predictor of segment membership, with those in younger age groups more likely to be

in higher utilisation segments. Those aged 5–10 years or 11–15 years had a 96% lower relative risk of being in the highest utilisation segment compared with the lowest utilisation segment, than those aged under 1 year after adjustment (RRR 95% CI: 4% to 5% and 3% to 4%, respectively). After adjustment, those of Asian or Asian British ethnicity were more likely to be in higher utilisation segments than those of other ethnicities, and with those of white ethnicity significantly less likely to be in each of the five higher utilisation segments.

A strong association was seen with higher levels of deprivation and membership of higher utilisation segments. Those with at least one LTC were 14.2 times (RRR 95% CI: 13.4 to 15.1) more likely to be in segment VI than segment I, 7.0 (RRR 95% CI: 6.7 to 7.4) times more likely to be in segment IV and 4.8 (RRR 95% CI: 4.6 to 5.1) times more likely to be in segment V, compared

**Table 4** Adjusted multinomial logistic regression of factors associated with segment assignment for 269 248 participants with non-missing data

| Segment | | RRR | SE | Z-score | P value | 95% CI |
|---|---|---|---|---|---|---|
| I: very low | Gender | Base outcomes | | | | |
| | Female | | | | | |
| | Age (years) | | | | | |
| | Under 1 | | | | | |
| | Ethnicity | | | | | |
| | Asian or Asian British | | | | | |
| | IMD quintile | | | | | |
| | 1 (most deprived) | | | | | |
| | LTCs | | | | | |
| | None | | | | | |
| II: low; GP | Gender | | | | | |
| | Male | 0.93 | 0.01 | −7.29 | <0.001 | 0.91 to 0.95 |
| | Age (years) | | | | | |
| | 1–2 | 0.90 | 0.04 | −2.75 | 0.01 | 0.83 to 0.97 |
| | 3–4 | 0.65 | 0.03 | −11.13 | <0.001 | 0.60 to 0.70 |
| | 5–10 | 0.39 | 0.01 | −26.36 | <0.001 | 0.37 to 0.42 |
| | 11–15 | 0.30 | 0.01 | −33.68 | <0.001 | 0.28 to 0.32 |
| | Ethnicity | | | | | |
| | White | 0.74 | 0.01 | −22.10 | <0.001 | 0.73 to 0.76 |
| | Mixed | 1.06 | 0.02 | 3.20 | <0.001 | 1.02 to 1.09 |
| | Other ethnic groups | 0.83 | 0.01 | −10.52 | <0.001 | 0.80 to 0.86 |
| | Black or Black British | 0.85 | 0.02 | −8.10 | <0.001 | 0.82 to 0.88 |
| | IMD quintile | | | | | |
| | 2 | 0.89 | 0.01 | −7.45 | <0.001 | 0.86 to 0.91 |
| | 3 | 0.83 | 0.01 | −11.27 | <0.001 | 0.80 to 0.85 |
| | 4 | 0.78 | 0.01 | −13.33 | <0.001 | 0.75 to 0.81 |
| | 5 (least deprived) | 0.82 | 0.02 | −8.10 | <0.001 | 0.79 to 0.86 |
| | LTCs | | | | | |
| | One or more | 2.29 | 0.05 | 35.17 | <0.001 | 2.19 to 2.40 |
| | Constant | 2.96 | 0.11 | 28.26 | <0.001 | 2.74 to 3.19 |
| III: moderate; GP | Gender | | | | <0.001 | |
| | Male | 0.91 | 0.01 | −8.23 | <0.001 | 0.88 to 0.93 |
| | Age (years) | | | | <0.001 | |
| | 1–2 | 0.50 | 0.02 | −18.60 | <0.001 | 0.46 to 0.54 |
| | 3–4 | 0.26 | 0.01 | −36.82 | <0.001 | 0.24 to 0.28 |
| | 5–10 | 0.09 | 0.00 | −73.43 | <0.001 | 0.08 to 0.09 |
| | 11–15 | 0.07 | 0.00 | −78.84 | <0.001 | 0.06 to 0.07 |
| | Ethnicity | | | | <0.001 | |
| | White | 0.49 | 0.01 | −45.98 | <0.001 | 0.48 to 0.51 |
| | Mixed | 0.70 | 0.01 | −18.41 | <0.001 | 0.67 to 0.73 |
| | Other ethnic groups | 0.67 | 0.01 | −20.45 | <0.001 | 0.64 to 0.70 |
| | Black or black British | 0.63 | 0.01 | −19.98 | <0.001 | 0.60 to 0.66 |
| | IMD quintile | | | | <0.001 | |
| | 2 | 0.88 | 0.02 | −6.80 | <0.001 | 0.85 to 0.91 |

Continued

| Table 4 Continued | | | | | | |
|---|---|---|---|---|---|---|
| Segment | | RRR | SE | Z-score | P value | 95% CI |
| | 3 | 0.77 | 0.01 | −13.37 | <0.001 | 0.74 to 0.80 |
| | 4 | 0.67 | 0.01 | −18.28 | <0.001 | 0.64 to 0.70 |
| | 5 (least deprived) | 0.82 | 0.02 | −7.37 | <0.001 | 0.77 to 0.86 |
| | LTCs | | | | <0.001 | |
| | One or more | 5.80 | 0.13 | 75.59 | <0.001 | 5.54 to 6.07 |
| | Constant | 8.00 | 0.29 | 56.48 | <0.001 | 7.44 to 8.59 |
| IV: moderate; GP, OP and elective | Gender | | | | <0.001 | |
| | Male | 1.08 | 0.02 | 5.00 | <0.001 | 1.05 to 1.11 |
| | Age (years) | | | | <0.001 | |
| | 1–2 | 0.57 | 0.03 | −11.84 | <0.001 | 0.51 to 0.62 |
| | 3–4 | 0.41 | 0.02 | −19.52 | <0.001 | 0.37 to 0.44 |
| | 5–10 | 0.20 | 0.01 | −38.29 | <0.001 | 0.19 to 0.22 |
| | 11–15 | 0.21 | 0.01 | −37.12 | <0.001 | 0.19 to 0.23 |
| | Ethnicity | | | | <0.001 | |
| | White | 0.78 | 0.02 | −12.86 | <0.001 | 0.75 to 0.81 |
| | Mixed | 1.05 | 0.03 | 1.96 | 0.05 | 1.00 to 1.10 |
| | Other ethnic groups | 0.96 | 0.02 | −1.66 | 0.10 | 0.92 to 1.01 |
| | Black or black British | 0.96 | 0.03 | −1.30 | 0.19 | 0.91 to 1.02 |
| | IMD quintile | | | | | |
| | 2 | 0.73 | 0.02 | −14.28 | <0.001 | 0.70 to 0.77 |
| | 3 | 0.64 | 0.01 | −19.54 | <0.001 | 0.61 to 0.67 |
| | 4 | 0.55 | 0.01 | −22.59 | <0.001 | 0.52 to 0.58 |
| | 5 (least deprived) | 0.69 | 0.02 | −11.10 | <0.001 | 0.65 to 0.74 |
| | LTCs | | | | <0.001 | |
| | One or more | 7.01 | 0.17 | 78.05 | <0.001 | 6.67 to 7.36 |
| | Constant | 1.53 | 0.07 | 9.31 | <0.001 | 1.40 to 1.68 |
| V: moderate; GP and emergency | Gender | | | | <0.001 | |
| | Male | 1.06 | 0.01 | 4.13 | <0.001 | 1.03 to 1.08 |
| | Age (years) | | | | <0.001 | |
| | 1–2 | 0.39 | 0.01 | −25.17 | <0.001 | 0.36 to 0.42 |
| | 3–4 | 0.18 | 0.01 | −46.40 | <0.001 | 0.17 to 0.20 |
| | 5–10 | 0.06 | 0.00 | −84.94 | <0.001 | 0.06 to 0.06 |
| | 11–15 | 0.04 | 0.00 | −92.34 | <0.001 | 0.04 to 0.04 |
| | Ethnicity | | | | <0.001 | |
| | White | 0.66 | 0.01 | −24.96 | <0.001 | 0.63 to 0.68 |
| | Mixed | 0.97 | 0.02 | −1.70 | 0.09 | 0.93 to 1.01 |
| | Other ethnic groups | 0.81 | 0.02 | −9.64 | <0.001 | 0.78 to 0.85 |
| | Black or black British | 0.73 | 0.02 | −12.05 | <0.001 | 0.70 to 0.77 |
| | IMD quintile | | | | <0.001 | |
| | 2 | 0.80 | 0.02 | −11.67 | <0.001 | 0.77 to 0.83 |
| | 3 | 0.69 | 0.01 | −17.75 | <0.001 | 0.67 to 0.72 |
| | 4 | 0.56 | 0.01 | −24.17 | <0.001 | 0.54 to 0.59 |
| | 5 (least deprived) | 0.59 | 0.02 | −16.85 | <0.001 | 0.56 to 0.63 |

Continued

| Segment | | RRR | SE | Z-score | P value | 95% CI |
|---|---|---|---|---|---|---|
| | LTCs | | | | <0.001 | |
| | One or more | 4.84 | 0.12 | 61.75 | <0.001 | 4.61 to 5.09 |
| | Constant | 7.59 | 0.28 | 54.18 | <0.001 | 7.06 to 8.17 |
| VI: high; all services | Gender | | | | | |
| | Male | 1.17 | 0.02 | 7.55 | <0.001 | 1.12 to 1.21 |
| | Age (years) | | | | <0.001 | |
| | 1–2 | 0.34 | 0.02 | −23.77 | <0.001 | 0.31 to 0.37 |
| | 3–4 | 0.13 | 0.01 | −42.88 | <0.001 | 0.12 to 0.15 |
| | 5–10 | 0.04 | 0.00 | −76.73 | <0.001 | 0.04 to 0.05 |
| | 11–15 | 0.04 | 0.00 | −77.55 | <0.001 | 0.03 to 0.04 |
| | Ethnicity | | | | <0.001 | |
| | White | 0.69 | 0.02 | −14.05 | <0.001 | 0.65 to 0.73 |
| | Mixed | 0.97 | 0.03 | −0.89 | 0.38 | 0.91 to 1.03 |
| | Other ethnic groups | 0.91 | 0.03 | −2.87 | <0.001 | 0.86 to 0.97 |
| | Black or black British | 0.85 | 0.03 | −4.20 | <0.001 | 0.79 to 0.92 |
| | IMD quintile | | | | <0.001 | |
| | 2 | 0.72 | 0.02 | −11.83 | <0.001 | 0.68 to 0.76 |
| | 3 | 0.58 | 0.02 | −17.84 | <0.001 | 0.55 to 0.62 |
| | 4 | 0.43 | 0.02 | −23.07 | <0.001 | 0.40 to 0.46 |
| | 5 (least deprived) | 0.50 | 0.02 | −14.11 | <0.001 | 0.46 to 0.55 |
| | LTCs | | | | <0.001 | |
| | One or more | 14.23 | 0.42 | 90.16 | <0.001 | 13.44 to 15.08 |
| | Constant | 2.55 | 0.12 | 19.84 | <0.001 | 2.32 to 2.79 |

GP, general practice; IMD, Index of Multiple Deprivation; LTCs, long-term conditions; OP, Outpatient; RRR, relative risk ratio.

with those without any LTC. However, the majority of CYP in all segments had no documented LTC, with the highest proportion (25.0%) seen in segment VI. A sensitivity analysis excluding ethnicity (due to 25.4% missing data) showed no marked differences in coefficients for the remaining variables, indicating that ethnicity was not a confounder of these relationships.

## DISCUSSION

Using a data-driven approach, we defined six segments of the CYP population in Northwest London each based on patterns of healthcare utilisation. To the best of our knowledge, this is the first application of segmentation approaches exclusive to the CYP population, including the whole population for a single geographic region. The highest utilisation group, representing less than 5% of the total population, accounted for over 50% of total emergency admissions and length of stay for the whole population. Age and deprivation were strong predictors of segment membership, with a general tendency toward lower utilisation patterns in older age groups and higher utilisation patterns in children living in areas of higher deprivation. Increased utilisation segments

were associated with higher total costs, with the two smallest segments, representing 13% of the population, accounting for almost two-thirds of costs.

Those with one or more LTCs were significantly more likely to be in higher utilisation segments than those without an LTC, but only one quarter of those in the high utilisation group had an LTC. Existing segmentation approaches applied to the adult population have focused on defining segments based on demographics and LTCs, with some incorporating healthcare utilisation.[8] Although the two approaches are closely linked in adult populations, our findings suggest that they are less tightly bound in the CYP population, with three-quarters in the high utilisation segment not having a recorded LTC. As a result, segmentation based on demographic factors and clinical comorbidities may fail to identify patients with higher rates of healthcare utilisation and vice versa, indicating a less well-defined trajectory of service use in children than in adults. Although this may make identification of high-need individuals using demographics and clinical factors more difficult, it suggests the potential for a greater impact of early interventions on healthcare utilisation and long-term outcomes.

The variation in healthcare utilisation with age is in line with previous studies showing significantly lower rates of primary and secondary care attendances with increasing age, and the greatest use of primary care and emergency admissions in infants.[25 26] Our study applied clustering to the whole CYP population and stratified clustering approaches to individual age groups might result in variation in the number and character of segments between age groups.

## Implications for policy and practice

Our study demonstrates the feasibility and potential benefits of using a population segmentation approach to detect utilisation for CYP population health rather than reactive, condition-specific strategies which have often been used historically. More widespread use of segmentation models for CYP could lead to more effective and efficient strategies tailored toward local contexts. In addition, our findings provide robust empirical evidence of the ways in which segmentation and appropriate policy responses for each segment differ between CYP and adult populations. Commissioners and policy-makers should ensure that there are separate local and national policies in place, which recognise and take these differences between children and adults into account.

In the Northwest London context, we are currently using segmentation techniques to target interventions for two specific groups of children: infants, where A&E attendances and primary care utilisation have increased significantly in the last 10 years[25] and those pre-schoolers who are frequent A&E attenders who make up approximately 7% of attendees but account for 40% of the workload.[27]

Our findings of higher utilisation patterns in certain groups, such as Asian ethnic groups and those resident in more deprived areas highlight potential inequalities in health and access to services. Lack of access to one service can lead to a corresponding increase in utilisation of other services, with a recent study indicating a lower ratio of GP and outpatient care versus emergency and inpatient care in children from more deprived areas, and significantly higher primary care use in children from Asian ethnic groups.[28] In children, the factors impacting on healthcare utilisation are complex and multifactorial, involving psychosocial and mental health problems, school absence and parental anxiety and depression—which could not be incorporated in this study.[29 30] Further work is needed to identify whether the results of our study indicate these populations to have greater and/or unmet health needs.

## Implications for segmentation approaches in children

There are broader implications of these findings on the value of segmentation approaches in children. A limitation to the use of healthcare utilisation is that it may not accurately reflect healthcare need. A service may be indicated, but remain under-used, or may not be indicated, but used frequently, influenced by factors, including proximity and access to services, to the requirement and ability to pay.[31]

Furthermore, extending from describing utilisation patterns to prediction of those with higher needs in order to intervene can unintentionally reinforce pre-existing inequities in access to healthcare. Obermeyer et al[32] identified significant racial bias in an algorithm identifying need based on health costs, with black patients showing higher rates of uncontrolled comorbid illness than white patients for a given risk score, due to misattribution of reduced access to care implying lower need.[32] Many of the factors related to utilisation patterns are unlikely to be captured in quantitative data, and further qualitative evaluations exploring child and parental factors related to utilisation patterns would be beneficial.[33]

Unlike with LTC-based segmentation, in which the conditions necessary for segment assignment are by definition 'chronic', healthcare utilisation will fluctuate over time. Identifying the trajectories of movement between segments might allow for early recognition and targeted intervention to reduce the risk of transition into a higher utilisation segment. In CYP, healthcare utilisation patterns are unlikely to be static, resulting in frequent movement between utilisation segments. This is supported by our finding that assignment to lower utilisation segments is more likely with increasing age and previous research showing that fewer than one-third of children accounting for the top 5% of healthcare costs in England were in the same group the following year.[12] The COVID-19 pandemic is likely to alter utilisation patterns both in the short and long term, with rates of primary and secondary care emergency attendance and paediatric intensive care unit admissions found to have fallen significantly in Scotland.[34] Further research is needed to understand the transitions between utilisation segments and the extent to which these transitions can be predicted.

A core ambition of segmentation approaches is to identify those at greatest risk and intervene pre-emptively, but evidence for the success of such interventions is mixed, and focused on adult populations. An intervention based around improving care coordination in the USA for adults with heart failure at high risk of secondary care utilisation demonstrated a reduction in A&E attendances and primary care visits, but not hospitalisations.[35] There is a risk in observational studies that by sampling high utilisers and following up over time, regression to the mean occurs, with overestimation of the intervention effect. A recent randomised, controlled trial in the USA targeting 'superutilisers' with care coordination following discharge found no significant difference in readmission rates compared with usual care.[36] Within the child population, evidence is more limited. A population health management programme in the USA targeted at under 21s in the top 10% of Medicaid expenditure showed a reduction in costs for those enrolled, of $378 per month per participant in the first year, but critically, showed no significant difference to a control group.[37]

## Limitations

The use of Discover data represents a population in a single, localised geography and the findings may not be representative of the CYP population at a national or international level and certain groups may be underrepresented or overrepresented. The population of Northwest London includes four of the ten most ethnically diverse local authorities in England, with only 29.6% of white ethnic backgrounds (of those with non-missing data on ethnicity), compared with 86.0% in England and Wales in 2011.[38 39] There are also relatively more people resident in more deprived areas compared with the national average, with only 7.1% in the least deprived quintile of deprivation. Discover provides comprehensive population coverage: 98% of the UK population are estimated to be registered to a GP, and 95% of those registered in Northwest London are captured in Discover.[13] However, it is likely that some of the most vulnerable groups, such as asylum seekers, refugees, travellers and homeless people, who may be disproportionate users of services such as emergency care, are not represented in these data.[40]

Use of linked routinely collected primary and secondary care datasets such as Discover, designed primarily for commissioning purposes and for direct clinical care, is still in their infancy for research purposes when compared with existing datasets such as the UK-based Clinical Practice Research Datalink (CPRD) and Hospital Episode Statistics. Data on LTCs were collected from the primary care record, which relies on routine input from GPs. The validity of diagnostic coding in CPRD has been extensively investigated, with chronic diseases shown to be generally well recorded.[41] Although no direct research has investigated coding validity in Discover, previous work has suggested a comparable prevalence of disease in Discover with national rates for those diseases defined by the Quality and Outcomes Framework (QOF).[13] However, many of the conditions relevant to LTCs in children do not form part of QOF, and so may be more likely to be incomplete, with evidence suggesting neurodevelopmental disorders in children, for example, are under-reported in the primary care record.[42] Obesity was prevalent in only 1.9% in our study, compared with a London average of 10.0% in reception children and 23.7% in year 6 children, suggesting LTCs in our analysis may be underestimated.[43] Total cost data were lower than national reported figures, with a recent report identifying annual costs of £113 and £380 in under 18s for primary and secondary care services, respectively, in 2015–2016, compared with £92 and £280 in our study.[44] Cost data will also only measure part of the impact of service use, with quality of life and non-health-related outcomes, including education, important metrics that were outside the scope of the current study.

The segmentation method used here was primarily data driven; however, when using unsupervised clustering, there are no well-established criteria for choosing the optimal number of segments and a degree of subjectivity is inevitable.[9] Although metrics such as the C-H and D-B scores, used here, give an indication as to the separation of clusters, there is no single metric which can determine the 'best' model, and different metrics may provide conflicting optima. Selection of the seven healthcare utilisation variables also impacts the cluster assignment; our model may weight inpatient admissions to a higher degree than other attendances, by including separately both elective and emergency admissions and lengths of stay. Different weightings can be applied to variables within the clustering process, to increase or decrease their relative importance, but there is no ground truth from which to evaluate whether clusters defined in this way are better indicators of the utilisation in the population.

The defined segments are also not agnostic to the choice of clustering algorithm. Using a hierarchical approach with Ward's method on a subset of data, clusters were descriptively similar in terms of relative differences in utilisation, but with only an 85% match in terms of homogeneity and completeness. A disadvantage of k-means clustering is that changes to the number of segments can lead to changes in inclusion of previous clusters, although the six clusters appeared stable with respect to five or seven clusters, with new clusters in each case split from one or two larger clusters, lending the transition between k-means clustering outputs a quasi-hierarchical character. Use of hierarchical clustering would ensure the stability of clusters over a range of total clusters but was not computationally feasible for the entire dataset.

## CONCLUSION

As integrated care services focus on population health and preventive care, it is imperative that approaches are developed to identify those CYP at a greater risk. In this analysis, we describe how use of segmentation in the CYP population can identify distinct groups based on different patterns of utilisation of services. Demographics and the presence of LTCs may not explain the variation in healthcare utilisation in children to the same extent as in adults. This suggests a less well-defined trajectory of healthcare utilisation in children, which might indicate greater potential for the positive impact of early interventions. Further research is needed to understand both how CYP transition between utilisation segments, and whether additional factors available in routine EHRs can explain variation in utilisation, to enable early identification and intervention.

**Author affiliations**
[1]Department of Primary Care and Public Health, Imperial College London, London, UK
[2]National Institute for Health Research Applied Research Collaboration Northwest London, Imperial College London, London, UK
[3]Centre for Mathematics of Precision Healthcare, Imperial College London, London, UK
[4]Department of Mathematics, Imperial College London, London, UK
[5]North West London Collaboration of Clinical Commissioning Groups, London, UK

**Acknowledgements** Many thanks to Eamon O'Doherty for his support with Discover access and data queries.

**Contributors** TB, JC, MBlair and DSH conceived the study. TB, JC, TW, RM, KS, MBarahona, MBlair and DSH were involved in the design of the work and interpretation of the findings. TB conducted the cohort extraction and statistical analysis on Discover database. TB and JC conducted the cluster analysis. TB wrote the first draft of the manuscript. TB, JC, TW, RM, KS, MBarahona, MBlair and DSH contributed to critical revisions of the manuscript and approved the final submission. TB is the guarantor and accepts full responsibility for the finished work and the conduct of the study, had access to the data, and controlled the decision to publish.

**Funding** TB is supported by a National Institute for Health Research (NIHR) Academic Clinical Fellowship and by the NIHR Applied Research Collaboration (ARC) programme for Northwest London. JC and M Barahona acknowledge support from EPSRC (grant number: EP/N014529/1) supporting the EPSRC Centre for Mathematics of Precision Healthcare. JC is also supported by the Wellcome Trust (grant number: 215938/Z/19/Z). TW is supported by the NIHR ARC programme for Northwest London. The views expressed in this publication are those of the authors and not necessarily those of the NHS, the NIHR or the Department of Health.

**Competing interests** None declared.

**Patient consent for publication** Not applicable.

**Ethics approval** Access to the Discover dataset was granted by the Northwest London Data Access Sub-group on 17 September 2020. Ethical approval was not required for this study as work was carried out for the purposes of service evaluation.

**Provenance and peer review** Not commissioned; externally peer reviewed.

**Data availability statement** Data for this study is not publicly available, but access to pseudonymised data may be accessible for research purposes or service evaluation through application to the North West London Data Access Sub-group.

**ORCID iDs**
Thomas Beaney http://orcid.org/0000-0001-9709-7264
Jonathan Clarke http://orcid.org/0000-0003-1495-7746
Mauricio Barahona http://orcid.org/0000-0002-1089-5675

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
