## [Reviewer comments · BMJ Open]

ARTICLE DETAILS

TITLE (PROVISIONAL)	Patterns of healthcare utilisation in children and young people: a retrospective cohort study using routinely collected healthcare data in North West London
AUTHORS	Beaney, Thomas; Clarke, Jonathan; Woodcock, Thomas; McCarthy, Rachel; Saravanakumar, Kavitha; Barahona, Mauricio; Blair, Mitch; Hargreaves, Dougal

VERSION 1 – REVIEW

REVIEWER	Kim, Min Sun Seoul National University Hospital, Department of Pediatrics
REVIEW RETURNED	08-May-2021

GENERAL COMMENTS	I think this paper is an important study to examine how the segmentation method can be used in the pediatric population. Considering that unlike in the adult population, a large number of children do not have chronic conditions, this is the basic data that helps to study methods in public health that can better find the factors that determine health utilization in the future.
--

REVIEWER	Jensen, Kristin M University of Colorado Health, Pediatrics and Internal Medicine
REVIEW RETURNED	10-May-2021

GENERAL COMMENTS	This is an important study and a very strong manuscript. My suggestions aim to clarify the manuscript for the readers. - Please consider re-orienting the readers to the fact that you applied segmentation by utilization patterns in the abstract and in the early parts of the results and discussion, as there was enough discussion about the different ways to segment that it wasn't immediately clear what was chosen.- When discussing findings, please consider presenting the findings of factors associated with higher utilization in the same direction (not increased in one case and decreased in the other) (page 15 lines 12-16).- If word limit allows, I would welcome deeper discussion surrounding whether some of the utilization differences by ages are expected vs. unexpected (ie - well child visits vs acute visits), as well as expansion of the concepts discussed on page 16, lower paragraph from your work that appears to be also under review.- Do the authors have thoughts regarding identifying and following health outcomes in place of utilization or cost? Overall, well done. I look forward to the finished product.
--

REVIEWER	Wood, Sophie
-----------------	--------------

	Cardiff University, CASCADE
REVIEW RETURNED	07-Oct-2021

GENERAL COMMENTS	Summary This is a well written paper using segmentation methods to identify patterns of healthcare usage for children and young people in North West London. The authors could reflect further on the implications of their study on policy, practice, and the children they serve. The finding that only one quarter of the higher utilisation group had a long-term health condition is important. This in combination with the finding that increased deprivation and being part of an ethnic minority group was more likely to predict membership to the higher utilisation segment, suggests issues of inequality which the authors do not reflect upon. Overall, the paper is an important contribution to child healthcare utilisation research and the use of routinely collected data. Therefore, I would suggest the authors to revise and re-submit. Major issues The Introduction is missing the “so what” factor. I can understand why this research is important, but you need to tell your reader. Who does this research benefit and how? Why is there a pressing need for the research? The appendix is far too long- be more selective about what you are including in your supplementary material. You also copy chunks of text from the main paper and include them in the appendix. Please don't duplicate text like this. Discussion – You need to reflect on how representative your sample is to the rest of England in terms of age, ethnicity, % of LTCs and deprivation. Discussion- you need to reflect more on what your findings mean for policy and practice. Minor issues Abstract: Participants, line 32 – Don't abbreviate NW (or abbreviate it above). Abstract: Participants, line 35 – include total sample size. Abstract: Primary and secondary outcome measures, line 39 – I appreciate the word count is limited, however it would be helpful for the reader if you summarised what the healthcare variables were e.g. ranging from general practice attendances to emergency care. Abstract: Conclusion- it is a bit contradictory. I would remove the word “distinct” from the first sentence. Introduction – p5, line 14. Cut the paragraph at “individual patients.” And say what the paper is concerned with (one sentence). Introduction- p5, second paragraph- Briefly explain what segmentation approaches are. I would add this just before you talk about adult population segmentation approaches. Introduction - p6, line 10 (also relates to other references of unpublished work) – You need to include the date. Can you also check with the editor that this is how they would want you to cite it? Seems awfully long. Introduction - p6, line 12 – I think you need to emphasise either here or the next paragraph, that this is why you are also going to consider sex, ethnicity and deprivation. Method- p6, line 55- What is the matching rate to secondary care? Method- p7, line 8 – needs rewording. “currently” registered is unclear. I would say... All CYP were included in the cohort if they
---

	were aged between 0 to 15 and were registered with a GP on the 01/03/2019 and had 365 days of continuous follow up until 29/02/2020. Method p7, line 15 – Does this mean children who died had a different study start date? How could this effect your results if you are measuring different time periods for these children? Results p9, line 53 – Add a “d” to age Results -p11, line 32 (but applies elsewhere)- I think you need to call the appendix “supplementary material”, and “p10” in your word document may not be the same in the published version, so use an alternative reference. Results p13, line 5- “Total healthcare costs for the cohort amounted to £140 million over the year.” It would be helpful to have some context either here or in the discussion. Is this expected? What is the English average for this age group? Results p13 line 37 – Could you re-do the segmentation analysis with different age bands as under 1s are likely to have different healthcare usage to 15-year-olds (as you have found), so this could be diluting some of your results. If you can’t re-do, then reflect on this in the discussion. Results p14 line 42 – Pseudo R-squared should only be used to compare models using the same data predicting the same outcome. It shouldn’t be used in the same way as R-squared, so I would therefore remove this sentence and interpretation. Conclusion p20, line 8 – “demographics and presence of long-term conditions may not adequately explain the variation in healthcare utilisation in children”. Could you tone this down? Perhaps if you slit your sample into age groups you would find a better explanation. Table 1 – The number of participants in your title is different to your total at the bottom of the table. Fig 1. – Your legends are too small to read.
--	--

VERSION 1 – AUTHOR RESPONSE

Reviewer 1

Comments to the Author:

I think this paper is an important study to examine how the segmentation method can be used in the pediatric population. Considering that unlike in the adult population, a large number of children do not have chronic conditions, this is the basic data that helps to study methods in public health that can better find the factors that determine health utilization in the future.

Author reply: Thank you for your review and positive comments on the paper.

Reviewer 2

Comments to the Author:

This is an important study and a very strong manuscript.

Author reply: thank you for your feedback on the paper and helpful comments which we have incorporated

My suggestions aim to clarify the manuscript for the readers.

- Please consider re-orienting the readers to the fact that you applied segmentation by utilization patterns in the abstract and in the early parts of the results and discussion, as there was enough discussion about the different ways to segment that it wasn't immediately clear what was chosen.

Author reply: we have added text to the abstract results to clarify segmentation was based on healthcare utilisation: “378,309 children were grouped into six segments based on healthcare utilisation”.

We have also added to the first sentence of the 'Model selection' section of results that the clustering was run "on the seven healthcare utilisation variables". We have also modified the first sentence of the discussion to clarify the segments were "based on" healthcare utilisation.

- When discussing findings, please consider presenting the findings of factors associated with higher utilization in the same direction (not increased in one case and decreased in the other) (page 15 lines 12-16).

Author reply: thank you for this suggestion. We have made some changes to the wording of the two paragraphs discussing the regression results to focus on the factors associated with higher utilisation segments where possible, although due to the use of particular baseline groups, this is not possible in every case.

- If word limit allows, I would welcome deeper discussion surrounding whether some of the utilization differences by ages are expected vs. unexpected (ie - well child visits vs acute visits), as well as expansion of the concepts discussed on page 16, lower paragraph from your work that appears to be also under review.

Author reply: we have added a new paragraph (3rd paragraph of discussion, p13) to contextualise the findings of age with previous research findings. We discuss ethnicity/deprivation differences in a newly added 'Implications for policy and practice' section of discussion on p15, along with the implications and need for further research in this area.

- Do the authors have thoughts regarding identifying and following health outcomes in place of utilization or cost?

Author reply: thanks for raising this. Although we did include children who died over the study period, there were small numbers in the whole population preventing a detailed exploration of this group. Other health outcomes, particularly those impacting quality of life and non-health related outcomes are important to consider, but outside the scope of data available in the EHR - we have commented on this in the limitations section:

"Cost data will also only measure part of the impact of service use, with quality of life and non-health related outcomes, including education, important metrics that were outside the scope of the current study."

Overall, well done. I look forward to the finished product.

Reviewer 3

Comments to the Author:

Summary

This is a well written paper using segmentation methods to identify patterns of healthcare usage for children and young people in North West London. The authors could reflect further on the implications of their study on policy, practice, and the children they serve. The finding that only one quarter of the higher utilisation group had a long-term health condition is important. This in combination with the finding that increased deprivation and being part of an ethnic minority group was more likely to predict membership to the higher utilisation segment, suggests issues of inequality which the authors do not reflect upon. Overall, the paper is an important contribution to child healthcare utilisation research and the use of routinely collected data. Therefore, I would suggest the authors to revise and re-submit.

Author reply: thank you for your thorough review and helpful comments. We have made changes to the manuscript which we hope now clarify our message, including addressing the comment above regarding inequalities in utilisation by ethnicity and deprivation.

Major issues

The Introduction is missing the "so what" factor. I can understand why this research is important, but you need to tell your reader. Who does this research benefit and how? Why is there a pressing need for the research?

Author reply: thank you for this important point, and we have added additional text in the introduction to clarify the relevance of the work. Specifically, much of the existing research in population health,

and in segmentation methods are focussed on the adult population, which is likely to be different to the CYP population and so dedicated research and development of approaches are needed. We have made changes throughout the introduction, including additional sentences to be explicit about the need for child-focussed research. At the end of intro, paragraph 1: “highlighting the need for population health approaches tailored towards this population” and intro, paragraph 2: “Existing work and development of segmentation methods have focussed on adults, who may represent a very different population to that of CYP in terms of chronic disease burden and patterns of healthcare utilisation, indicating a need for research focussed on the child population.” We have also clarified the aims in the final sentence of the introduction: “We then describe the population characteristics of the resulting segments, by age, sex, ethnicity, LTC burden and deprivation in order to identify population groups associated with different patterns of healthcare utilisation.”

The appendix is far too long- be more selective about what you are including in your supplementary material. You also copy chunks of text from the main paper and include them in the appendix. Please don't duplicate text like this.

Author reply: we have removed the first three figures from the appendix and cut the section on the cohort design, where the key points are already in the manuscript as you have highlighted. However, we feel the remaining figures are important in presenting our approach to the clustering, and the purpose of the appendix is optional for those readers with an interest in being able to replicate the methods, which would not be possible to do given the word count allowed for the manuscript. In order to provide further detail from the manuscript, it has been necessary to repeat a few sentences from the main paper, along with additional description – we have now added a note at the start of the appendix to acknowledge this to be the case. We do not think this should be an issue, as it is common practice in many journals, but if the editors prefer, we can discuss removing these sections from the appendix.

Discussion – You need to reflect on how representative your sample is to the rest of England in terms of age, ethnicity, % of LTCs and deprivation.

Author reply: we did comment in the limitations section of discussion on the diverse ethnic population in NW London compared with the rest of the UK. We have added additional text to the first paragraph of limitations (p17) referencing the proportion of white ethnicities nationally to make clear the comparison with NWL, and have discussed the relatively high proportion of the population from more deprived areas. Given our focus is on a relatively narrow age group, and the LTCs used here may be different to those using other chronic disease categories, we have not included a comparison with national age or LTCs estimates. The focus of the study is also on the internal comparisons within the study population, so the representativeness to national demographics is of less relevance than the extent to which particular population groups may or may not be represented, which we have commented on at the end of the first paragraph of limitations.

Discussion- you need to reflect more on what your findings mean for policy and practice.

Author reply: we have re-structured the discussion section, with a section on 'Implications for policy and practice'. We have incorporated additional text in this section on inequalities related to ethnic and socioeconomic deprivation, and have also integrated previous text on how segmentation methods are being used in NWL:

“Our study demonstrates the feasibility and potential benefits of using a population segmentation approach to detect utilisation for CYP population health rather than reactive, condition-specific strategies which have often been used historically. More widespread use of segmentation models for CYP could lead to more effective and efficient strategies tailored towards local contexts. In addition, our findings provide robust empirical evidence of the ways in which segmentation and appropriate policy responses for each segment differ between CYP and adult populations. Commissioners and policymakers should ensure that there are separate local and national policies in place which recognise and take these differences between children and adults into account.

In the Northwest London context, we are currently utilising segmentation techniques to target interventions for two specific groups of children: infants, where ED attendances and primary care utilisation have increased significantly in the last 10 years[25] and those pre-schoolers who are frequent ED attenders who make up approximately 7% of attendees but account for 40% of the workload.[30]

Our findings of higher utilisation patterns in certain groups, such as Asian ethnic groups and those resident in more deprived areas highlight potential inequalities in health and access to services. Lack of access to one service can lead to a corresponding increase in utilisation of other services, with a recent study indicating a lower ratio of GP and outpatient care vs emergency and inpatient care in children from more deprived areas, and significantly higher primary care use in children from Asian ethnic groups.[27] In children, the factors impacting on healthcare utilisation are complex and multifactorial, involving psychosocial and mental health problems, school absence and parental anxiety and depression – which could not be incorporated in this study.[28, 29] Further work is needed to identify whether the results of our study indicate these populations to have greater and/or unmet health needs.”

Minor issues

Abstract: Participants, line 32 – Don't abbreviate NW (or abbreviate it above).

Author reply: corrected throughout as Northwest

Abstract: Participants, line 35 – include total sample size.

Author reply: added

Abstract: Primary and secondary outcome measures, line 39 – I appreciate the word count is limited, however it would be helpful for the reader if you summarised what the healthcare variables were e.g. ranging from general practice attendances to emergency care.

Author reply: added

Abstract: Conclusion- it is a bit contradictory. I would remove the word “distinct” from the first sentence.

Author reply: thank you, we have changed this to “this article identifies six segments of healthcare utilisation in CYP.”

Introduction – p5, line 14. Cut the paragraph at “individual patients.” And say what the paper is concerned with (one sentence).

Author reply: we have made some changes to the opening paragraph in response to your first comment about the ‘so what’. We have not directly addressed the aim of the paper in the first paragraph, given the need to first introduce segmentation methods, but have highlighted the focus around addressing population health needs in the CYP population.

Introduction- p5, second paragraph- Briefly explain what segmentation approaches are. I would add this just before you talk about adult population segmentation approaches.

Author reply: we have amended this paragraph to give an overview of the aims of segmentations approaches first, and then discussed the application to the adult population.

Introduction - p6, line 10 (also relates to other references of unpublished work) – You need to include the date. Can you also check with the editor that this is how they would want you to cite it? Seems awfully long.

Author reply: these two unpublished works are now published, so both updated as references

Introduction - p6, line 12 – I think you need to emphasise either here or the next paragraph, that this is why you are also going to consider sex, ethnicity and deprivation.

Author reply: we agree this is important to highlight and have added to the end of this paragraph: “and that other factors, including those available in electronic health records (EHRs) such as sex, ethnicity and deprivation may also play a role.”

Method- p6, line 55- What is the matching rate to secondary care?

Author reply: Patients are matched based across datasets based on a pseudonymised NHS ID.

Secondary care data is sourced from SUS, and so patients will only be matched if they have had a

secondary care attendance. We have added a sentence into the healthcare utilisation section of the results, to state the total number and % matched with a secondary care record: “136,424 (36.1%) were matched with one or more records of any secondary care attendance, with 23.5% attending A&E and 21.2% attending outpatients at least once over the year.”

Method- p7, line 8 – needs rewording. “currently” registered is unclear. I would say... All CYP were included in the cohort if they were aged between 0 to 15 and were registered with a GP on the 01/03/2019 and had 365 days of continuous follow up until 29/02/2020.

Author reply: thank you, we agree this was unclear and have updated as you suggest

Method p7, line 15 – Does this mean children who died had a different study start date? How could this effect your results if you are measuring different time periods for these children?

Author reply: Those who died during the year from 01/03/2019 had a full year of follow-up included, and so an earliest allowed start date of 01/03/2018. This approach was chosen in order to ensure comparable time periods for those that died compared to those living and this should not impact on our results. An alternative would have been to annualise the proportion of the year of follow-up for those that died, but this would have added significant bias, as healthcare utilisation is likely to increase significantly immediately preceding death. We have added the earliest allowed start date to the methods, which was previously in the appendix.

Results p9, line 53 – Add a “d” to age

Author reply: corrected

Results -p11, line 32 (but applies elsewhere)- I think you need to call the appendix “supplementary material”, and “p10” in your word document may not be the same in the published version, so use an alternative reference.

Author reply: we have updated as ‘online supplemental file’. We are happy to update the final version of the paper with the correct page reference in the appendix.

Results p13, line 5- “Total healthcare costs for the cohort amounted to £140 million over the year.” It would be helpful to have some context either here or in the discussion. Is this expected? What is the English average for this age group?

Author reply: we have added the mean cost per child to the results, with the breakdown of primary vs secondary care costs. The aim here is to present the relative differences between segments, rather than to provide an overall estimation of costs, as we have not included all potential costs (eg prescription costs). We have added into the limitations a comparison of the costs compared to national estimates of healthcare costs in England.

Results p13 line 37 – Could you re-do the segmentation analysis with different age bands as under 1s are likely to have different healthcare usage to 15-year-olds (as you have found), so this could be diluting some of your results. If you can’t re-do, then reflect on this in the discussion.

Author reply: unfortunately, it is not possible to update the segmentation analyses at this stage stratified by different age bands. This would also add considerably to the complexity of the presentation and interpretation of the findings. However, this is an important consideration, and we have added in the new third paragraph of discussion (p14) where we talk about the age associations with utilisation:

“Our study applied clustering to the whole CYP population and stratified clustering approaches to individual age groups might result in variation in the number and character of segments between age groups.”

Results p14 line 42 – Pseudo R-squared should only be used to compare models using the same data predicting the same outcome. It shouldn’t be used in the same way as R-squared, so I would therefore remove this sentence and interpretation.

Author reply: thank you for raising this. We agree that it should not strictly be interpreted as a proportion of explained variation as for an R-squared, but that its low value does imply low explanatory power of the model. However, given this is a subjective assessment, and the key factor remains the significant proportion with no documented LTC, we have removed this sentence, and also made changes to the strengths and limitations section which had referenced this.

Conclusion p20, line 8 – “demographics and presence of long-term conditions may not adequately explain the variation in healthcare utilisation in children”. Could you tone this down? Perhaps if you slit your sample into age groups you would find a better explanation.

Author reply: we have amended this sentence to: “Demographics and presence of long-term conditions may not explain the variation in healthcare utilisation in children to the same extent as in adults.”

Table 1 – The number of participants in your title is different to your total at the bottom of the table.

Author reply: thanks for spotting this, the figure in the title has been updated to the correct figure as in the table denominator

Fig 1. – Your legends are too small to read.

Author reply: this has been updated

VERSION 2 – REVIEW

REVIEWER	Wood, Sophie Cardiff University, CASCADE
REVIEW RETURNED	24-Nov-2021
GENERAL COMMENTS	Thank you for taking the time to consider my revisions and make changes to the paper. This work is a great addition to the field and I look forward to seeing to published.